

# Impacts of air pollution and climate on materials in Athens, Greece

John Christodoulakis[1], Chris G. Tzanis[1], Costas A. Varotsos[1], Martin Ferm[2], Johan Tidblad[3]

[1]Climate Research Group, Division of Environmental Physics and Meteorology, Department of Physics, National and Kapodistrian University of Athens, University Campus Bldg. Phys. V, Athens 15784, Greece

[2]IVL Swedish Environmental Research Institute Ltd., P.O. Box 53021, SE-400 14 Gothenburg, Sweden

[3]Corrosion and Metals Research Institute, Drottning Kristinas väg 48, 114 28 Stockholm, Sweden

*Correspondence to*: Costas A. Varotsos (covar@phys.uoa.gr)

**Abstract.** For more than 10 years now the National and Kapodistrian University of Athens,, Greece, contributes to the UN/ECE ICP Materials programme for monitoring of the corrosion/soiling levels of different kind of materials due to

environmental air-quality parameters. In this paper we present the results obtained from the analysis of such observational data that were collected in Athens during the period 2003-2012. According to these results the corrosion/soiling of the particular exposed materials tend to decrease over the years, except for the case of copper. Based on this long experimental database applicable to multi-pollutant situation of the Athens basin we present dose response functions (DRFs) considering, that "*dose*" stands for the air pollutant concentration, "*response*" for the material mass loss (normally per annum) and the

"*function*" the relationship derived by the best statistical fit to the data.

## 1 Introduction

Climatic parameters and air pollutants are of major importance for the deterioration of many materials used in buildings and cultural monuments (Ferm et al., 2005, 2006; Varotsos et al., 2009; Tzanis et al., 2009a, 2011; Tidblad et al., 2012). These pollutants are mainly emitted by industrial and agricultural activities, as well as by the transport sector, and beyond their

effects on human health and ecosystems, they also contribute to the deterioration of cultural monuments both on the local scale and over long distances (Köhler et al., 2001; Ondov et al., 2006; Ebel et al., 2007; Tzanis et al., 2009b; Jacovides et al., 1994; Efstathiou et al., 2005; Varotsos et al., 2011, 2014; Reid et al., 1998; Chattopadhyay et al., 2012; Krapivin and Shutko, 2012; Merlaud et al., 2012; Xue et al., 2014; Monks et al., 2015). The world's cultural heritage is very diverse and costly to maintain. Repairing costs for deterioration of various materials due to air pollution, together with climatic parameters, are

huge (Doytchinov et al., 2011), while the damage to cultural objects endangers seriously the cultural heritage.

Effective policy making requires an adequate scientific basis to assess the effects of pollution and climate change on materials. In this context, the United Nations Economic Commission for Europe (UNECE) adopted the Convention on Long-range Transboundary Air Pollution (CLRTAP) to address the problems of air pollution. In the framework of the UNECE/CLRTAP the International Co-operative Programme on Effects on Materials including Historic and Cultural

Monuments (ICP Materials) was launched, in order to provide, among others, a scientific basis for the study of important



materials' degradation due to atmospheric pollution and climate parameters. The Athens, Greece with significant cultural heritage monuments (UNESCO Cultural Heritage site: Acropolis, Parthenon) has been involved in ICP Materials since 2002 as a targeted field exposure test site, participating also in the EU project MULTI-ASSESS (Model for multi pollutant impact and assessment of threshold levels for cultural heritage: http://www.corr-institute.se/multi-assess/web/page.aspx).

An important contribution to this effort is the development of dose response functions (DRFs) for particular materials. DRFs are relationships between the corrosion or soiling rates and the levels or loads of pollutants in combination with climatic parameters. The corrosion is mainly caused by chemical reactions on the material surface involving air pollutants (e.g., $SO_2$, $NO_x$ and $O_3$), while soiling is principally depicted as loss of reflectance (Watt et al., 2008). Concerning the latter, the incorporation of $PM_{10}$ concentration in the above mentioned relationship allows for the generation of empirical dose–

response functions for soiling (Brimblecombe and Grossi, 2005).

The DRFs are used for the assessment of pollution tolerable levels and to recommend target levels to be implemented in the future development of measures on urban air quality in order to minimise the pollution effects on historic and cultural objects. In addition, they can be used in sites where there are no experimental results in order to make estimations of corrosion/soiling rates. According to previous studies implemented in Athens, carbon steel has been proven that is the

material which suffers more from corrosion than the others exposed metals/alloys. On the contrary, copper is the most durable (Tzanis et al., 2011). Another study has revealed that the greatest part of the deposited particle mass is not water soluble, while in the water soluble part of it there is an unbalance between the cations and anions with the cations to surpass anions (Tzanis et al., 2009a).

In this study we present the most recent results from the UNECE/ICP Materials trend exposure programme 2011-2012

obtained in Athens, Greece test site, along with the corresponding measurements from previous exposure periods for comparison reasons. We also demonstrate the comparison between experimental results and theoretical corrosion/soiling estimations by employing the newly developed DRFs for the campaigns conducted in Athens, Greece.

## 2 Experimental

For the purpose of MULTI-ASSESS and UNECE ICP Materials trend exposure programmes, a station is installed in central

Athens, Greece (37°59′57′′ N, 23°43′59′′ E), since 2003. The main rack - field exposure site with exposure samples and the carousel on rack along with sheltered samples enclosed in a box under the rack, for the last exposure period, are shown in Fig. 1. Specimens of the materials carbon steel (6 samples), weathering steel (9 samples), zinc (6 samples), copper (3 samples), aluminium (3 samples), limestone (6 samples), and modern glass (1 sample) were installed on the main rack. The vast majority of the specimens were exposed in unsheltered positions, while the modern glass in sheltered position inside the

aluminium box with open bottom. The exposure time for modern glass and copper as well as for three samples of carbon steel, weathering steel, zinc and limestone was one year, while the rest samples are scheduled to be withdrawn in a later time.





The withdrawn specimens were sent to the responsible subcentres in Europe (see Table 1) for further analysis and evaluation of soiling or corrosion attack.

In particular, for the determination of multi-pollutant effects on materials, chemical analysis of the specimens was conducted and basic parameters as the weight change, mass loss, surface recession, haze, the total deposited mass of particles per

surface unit of glass (TP/S) were calculated. For comparison reasons, as also indicated in Introduction, the corrosion and soiling values for the exposure period 2011-2012 was complemented with the available data collected previously (2003-2004, 2005-2006 and 2008-2009) in the frame of MULTI-ASSESS and UNECE ICP Materials programmes, in which the Athens station has been involved.

In addition, the diffusive passive samplers for the surface air-pollutants ($SO_2$, $HNO_3$, HCOOH, $CH_3COOH$, HCl and HF)

measurements and the passive particle collector (aerosols) that were used (shown also in Fig. 1), were prepared at Swedish Environmental Research Institute (IVL). The samplers were mounted under a metal disc ca 2m above the ground in order to protect them from rain and direct sunshine and after the exposure, they were returned to IVL for analysis. The main aim of these measurements was to correlate the pollutants concentrations with the degradation rate of the exposed material specimens.

**3 Results and discussion**

As mentioned before, in order to study the corrosion of structural metals (copper, zinc, carbon and weathering steel), the parameters weight change and mass loss were evaluated. Figures 2-7 present the weight change and mass loss values obtained after the analysis of the exposed specimens. In these figures the experimental results of previous expositions are also presented. It should be mentioned that the presented values are the mean values obtained for the three specimens of each

structural metal exposed during the aforementioned exposure periods.

The parameter "weight change" describes the difference in specimen's mass after the exposure minus its initial mass. If the specimen was exposed under sheltered conditions this parameter is expected to be positive due to uptake processes (e.g. deposition) and the lack of any mass loss mechanism. In the case of unsheltered exposition, weight change can be positive or negative depending on the balance among uptake and loss mechanisms. According to the results obtained for the case of

copper (Fig. 2), mean weight change of samples exposed during 2011-2012 period is almost 1.5 times greater than that of the samples exposed during 2003-2004 (Tidblad et al., 2013).

The parameter "mass loss" expresses the difference in specimen's initial mass minus the specimen's mass after removing its corroded part. It should be mentioned here that both the weight change and mass loss parameters are affected by the run-off and the chemical composition of the corrosion layer (Horalek et al., 2005). The experimental results of the mass loss, for

copper, zinc and carbon steel, are presented in Figs. 3, 5 and 7, respectively. According to these results, mass loss of copper is shown to have increased since 2003-2004; however, this increase has been minimal (1.075 times greater). On the contrary, mass loss of zinc and carbon steel samples decreases continuously after the period 2005-2006. The greatest values of mass




loss for both materials were recorded for the case of Athens, Greece, during that period. Last results denote reduce of zinc mass loss of about 36% and reduce of carbon steel mass loss of about 55% since that period. The corrosion rates of carbon steel are shown to have decreased significantly during 2011-2012, possibly due to the reduced levels of $SO_2$ and $PM_{10}$ which have been measured. In addition, first results show that pollution has a significant effect on corrosion rate of weathering steel.

Mean mass loss of weathering steel samples during 2011-2012 exposition was evaluated to 82.8 g m$^{-2}$ (Tidblad et al., 2013). The carbon and weathering steel arises to be the most sensitive metals, among the exposed ones, to the mass loss, while copper is the most durable. That means that steel is the most sensitive material to the corrosion while copper suffered less by atmospheric corrosion. Considering climate change future projections it is expected an increase in temperature, relative humidity and precipitation (IPCC, 2013) factors which favour corrosion rate. However, corrosion rate is also affected by

pollutants levels which generally are decreasing. So the question "how much climate change affects materials corrosion?" needs very careful approach.

In the case of zinc samples, chemical analyses were performed to water solutions of the corrosion products. These solutions were analysed for inorganic acids, formate and acetate. The aim was the identification of corrosive media which affected metal surface. The results can not be used for quantitative analysis but they are useful for qualitative conclusions about the

15 substances which mainly corroded zinc samples (Tidblad et al., 2013). The analysis showed that chloride ions, water-soluble sulphate and nitrates are involved in the corrosion processes of the exposed zinc samples in Athens. No traces of formate and acetate were found.

For the evaluation of corrosion of limestone specimens exposed in unsheltered positions, surface recession, was calculated.

This parameter is defined by the formula $R = \dfrac{W_1 - W_0}{A \cdot \rho}$, where $W_0$ is sample's weight before the exposure, $W_1$ is sample's

weight after the exposure, A is the total surface area of sample and $\rho$ is the density of the limestone. The results of surface recession for the limestone specimens exposed, under unsheltered conditions, for one year are presented in Fig. 8 along with the same results obtained during previous exposure periods. Generally, the recession of limestone has decreased slightly after the period 2005-2006 due possible to the reduced pollution levels. It is also obvious from this figure that recession during last exposure period (2011-2012) is slightly higher than the previous one, perhaps due to a small increase in $NO_2$

concentration during this period.

Another material studied during this exposure period was modern glass. This one is not part of historic and cultural monuments but it is a material which is used widely in synchronous art as well as in other kind of modern constructions. In addition to that, modern glass is also an ideal material for soiling studies because it is transparent, flat, non-porous and chemically inert. Due to these properties modern glass does not affect particles deposition and accumulation (Lombardo et

al., 2010).

In order to evaluate soiling two parameters are investigated; the total deposited mass of particles per surface unit of glass (TP/S) in μg cm$^{-2}$ and haze defined as the ratio, expressed in percentage, of the diffuse to direct transmitted light. Modern glass samples were exposed under sheltered conditions during all exposure periods.





The obtained results for TP/S and haze are presented in Figs. 9 and 10, respectively. Regarding TP/S it shows a clear decreasing trend through the exposure periods. Maximum value was recorded during 2003-2004 and it is proven to be about 4 times greater than the next periods. Minimum value was recorded during 2011-2012 exposure period. The range of haze is similar for the exposure periods 2005-2006, 2008-2009 and 2011-2012 while the minimum value is presented for 2011-2012

and the maximum for 2003-2004.

The corrosion or soiling values presented above and environmental parameters mentioned in section 2, along with data from previous experimental campaigns, were analysed in order to develop the dose-response functions for corrosion and soiling for materials under study. The results for DRFs (for multi pollutant situation except for the case of weathering steel) based on data from all the ICP Materials test sites are presented below in Eqs. (1-6) (Kucera et al., 2005, 2007; Watt et al., 2008;

Verney-Carron and Lombardo, 2013) along with correlation coefficients $R^2$, Root Mean Square Deviations (RMSD) and Normalized Root Mean Square Deviations (NRMSD) between observed and predicted values for Athens, Greece. In addition to these, we present newly developed DRFs, Eqs. (7-10), along with the correlation coefficients $R^2$, RMSD and NRMSD between observed and new predicted values for carbon steel, zinc, limestone and modern glass for the case of Athens, Greece. The obtained values of these statistical parameters are given in Table 2. For copper and weathering steel the

available data were not adequate for developing new DRFs. All the presented below DRFs (Eqs 1, 2, 3, 4, 5, 7, 8, 9) are valid for one year exposure except for modern glass (Eqs. 6, 10) where t denotes the exposure duration in days. These DRFs are based on parameters already defined by UNECE/ICP Materials group and were obtained implementing nonlinear regression analysis for carbon steel, zinc and limestone and multiple linear regression for the modern glass case. It should be noted that the time factor in the new DRF for modern glass (Eq. 10) remained the same as in Eq. (6) (see Lombardo et al., 2010).

### Carbon steel

$$ML = 51 + 1.39[SO_2]^{0.6}Rh_{60}e^{f(T)} + 1.29Rain[H^+] + 0.593PM_{10} \hspace{2cm} (Eq.\ 1)$$

$f(T) = 0.15(T-10)$ when $T<10°C$ (Eq. 1.1), otherwise $f(T) = -0.054(T-10)$ (Eq. 1.2)

### Zinc

$$ML = 3.5 + 0.471[SO_2]^{0.22}e^{0.018Rh+f(T)} + 0.041Rain[H^+] + 1.37[HNO_3] \hspace{1cm} (Eq.2)$$

$f(T) = 0.062(T-10)$ when $T<10°C$ (Eq. 2.1), otherwise $f(T) = -0.021(T-10)$ (Eq. 2.2)

### Limestone

$$R = 4.0 + 0.0059[SO_2]Rh_{60} + 0.054Rain[H^+] + 0.078[HNO_3]Rh_{60} + 0.0258PM_{10} \hspace{1cm} (Eq.\ 3)$$

### Weathering steel (C<0.12%, Mn 0.3-0.8%, Si 0.25-0.7%, P 0.07-0.15%, S<0.04%, Cr 0.5-1.2%, Ni 0.3-0.6%, Cu 0.3-0.55%, Al<0.01%)

$$ML = 34[SO_2]^{0.13}e^{0.020Rh + f(T)} \hspace{3cm} (Eq.\ 4)$$



$f(T) = 0.059(T-10)$ when $T \leq 10°C$ (Eq. 4.1), otherwise $f(T) = -0.036(T-10)$ (Eq. 4.2)

*Copper*

$ML = 4.21 + 0.00201[SO_2]^{0.4}[O_3]Rh_{60}e^{f(T)} + 0.0878Rain[H^+]$ (Eq. 5)

$f(T) = 0.083(T-10)$ when $T \leq 10°C$ (Eq. 5.1), otherwise $f(T) = -0.032(T-10)$ (Eq. 5.2)

*Modern glass*

$H = (0.2215 [SO_2] + 0.1367 [NO_2] + 0.1092 PM_{10}) / (1 + (382/t)^{1.86})$ (Eq. 6)

*Carbon steel for Athens*

$ML = 10 + 0.012[SO_2]^{2.152}Rh_{60}e^{f(T)} + 1.29Rain[H^+] + 1.263PM_{10}$ (Eq. 7)

$f(T) = 0.15(T-10)$ when $T < 10°C$ (Eq. 7.1), otherwise $f(T) = -0.054(T-10)$ (Eq. 7.2)

*Zinc for Athens*

$ML = 3.5 + 0.004[SO_2]^{0.408}e^{0.082Rh+f(T)} + 0.041Rain[H^+] + 0.138[HNO_3]$ (Eq. 8)

$f(T) = 0.062(T-10)$ when $T < 10°C$ (Eq. 8.1), otherwise $f(T) = -0.021(T-10)$ (Eq. 8.2)

*Limestone for Athens*

$R = 4.0 + 0.002[SO_2]Rh_{60} + 0.054Rain[H^+] + 0.05[HNO_3]Rh_{60} + 0.106PM_{10}$ (Eq. 9)

*Modern glass for Athens*

$H = (0.204 [SO_2] + 0.016 [NO_2] + 0.319 PM_{10}) / (1+(382/t)^{1.86})$ (Eq. 10)

where

ML = mass loss by corrosion attack, $g\ m^{-2}$

R = surface recession, μm (absolute values)

t = exposure time, days

Rh = relative humidity, % - annual average

$Rh_{60}$ = Rh – 60 when Rh > 60, 0 otherwise

T = temperature, °C - annual average

$[SO_2]$ = concentration, $μg\ m^{-3}$ - annual average

$[O_3]$ = concentration, $μg\ m^{-3}$ - annual average

$[NO_2]$ = concentration, $μg\ m^{-3}$ - annual average

Rain = amount of precipitation, $mm\ year^{-1}$ - annual average





[HNO$_3$] = annual average concentration, $\mu g\ m^{-3}$

PM$_{10}$ = annual average concentration, $\mu g\ m^{-3}$

[H$^+$] = concentration, mg l$^{-1}$ - annual average. The unit for [H$^+$] is not the normal one (mol l$^{-1}$) used for this denomination and the relation between pH and [H$^+$] is therefore here [H$^+$] = 1007,97 $10^{-pH} \approx 10^{3-pH}$.

In the Figs. 11-15 we present the above DRFs' (for all the ICP Materials test sites ("ICP DRF") and for Athens ("Athens DRF")) results along with the experimental values ("Observed") obtained at Athens, Greece. For the case of weathering steel, the estimated mass loss is 100.6 g m$^{-2}$ while as mentioned before the observed value is 82.8 g m$^{-2}$. A general remark for the case of Athens is that the ICP DRFs results for the case of metals/alloys overestimate the corrosion levels while for

limestone and modern glass they underestimate corrosion/soiling levels for all the exposure periods. Specifically, in case of copper the overestimation is almost 17% for 2003-2004 period and almost 9% for the 2011-2012 period. In case of zinc the overestimated mass loss ranges from 8 to 47% for all exposure periods. Carbon steel mass loss is greater than the observed by 3 to 35% through all exposure periods, while the weathering steel's mass loss is estimated almost 22% greater than the observed one.

Limestone results reveal that DRFs estimations underestimate corrosion levels by 29 to 47%. In case of modern glass the observed haze is 4 to 34% greater than the estimated values for all the exposure periods except for the case of 2005-2006 where an overestimation of about 6% is noticed.

DRFs for Athens case present improved estimations. In particular, in case of zinc new DRF estimations underestimate mass loss by about 0% to 3% except for the case of 2008-2009 exposure period where an overestimation of 3% is noticed. In case

of carbon steel new estimations underestimate mass loss by about 1% for all exposure periods except for last one where an overestimation of 3% is noticed. New DRF estimations for limestone recession are between -14% (underestimation) to 10% (overestimation), while the estimated from Athens DRF modern glass haze differs from the observed values from -24 to 21%. This range of differences may indicate that for the Athens, Greece case the parameters used in DRF for the modern glass are not sufficient and more experimental data are needed in order to specify the factors which affect haze. In Fig. 16 are

presented the percentage contribution of each Athens DRF factor to the total corrosion/soiling of each material for all exposure periods.

## 4 Conclusions

According to the above mentioned results, all the exposed materials, except for copper, present reduced corrosion/soiling levels through the years. In case of copper, it presents almost 7% greater mass loss during the last exposure period than

during 2003-2004. According to DRFs O$_3$ is a parameter which affects copper mass loss, while it does not affect the rest materials. So a possible explanation to this could be the increased level of O$_3$ during 2011-2012 (23.7 $\mu g\ m^{-3}$) compared to 2003-2004 (19.7 $\mu g\ m^{-3}$). New developed DRFs for the particular case of Athens, Greece improve the obtained estimations



for corrosion and soiling of the materials under study. However, these DRFs will be re-evaluated when new data from the 2014-2015 exposure period are available.

## Acknowledgements

We gratefully acknowledge the Ministry of Health and all the participants in the MULTI-ASSESS and ICP Materials programmes.

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



**Table 1: Responsible subcentres for the evaluation of corrosion or soiling of the exposed materials for the period 2011-2012.**

| Material | Responsible subcentre |
|---|---|
| Carbon steel | SVUOM, Czech Republic |
| Weathering steel | CENIM/CSIC, Spain |
| Zinc | EMPA, Switzerland |
| Copper | KIMAB, Sweden |
| Limestone | BRE, Watford, UK |
| Modern glass | Univeristy Paris XII, LISA, France |

**Table 2: Correlation coefficients $R^2$, Root Mean Square Deviations (RMSD) and Normalized Root Mean Square Deviations (NRMSD) between observed and predicted values for Athens, Greece. The abbreviation "nss" declares not statistically significant value at 95% confidence interval while "ss" statistically significant value at 95% confidence interval.**

| Dose Response Function | | $R^2$ | RMSD | NRMSD (%) |
|---|---|---|---|---|
| Carbon steel | (Eq.1) | 0.972 (ss) | 12.57 | 19 |
| Carbon steel for Athens | (Eq.7) | 0.999 (ss) | 1.07 | 2 |
| Zinc | (Eq.2) | 0.581 (nss) | 2.01 | 80 |
| Zinc for Athens | (Eq.8) | 0.995 (ss) | 0.096 | 4 |
| Limestone | (Eq.3) | 0.556 (nss) | 3.79 | 230 |
| Limestone for Athens | (Eq.9) | 0.653 (ss) | 0.796 | 48 |
| Modern glass | (Eq.6) | 0.797 (nss) | 2.24 | 48 |
| Modern glass for Athens | (Eq.10) | 0.809 (ss) | 1.5 | 32 |



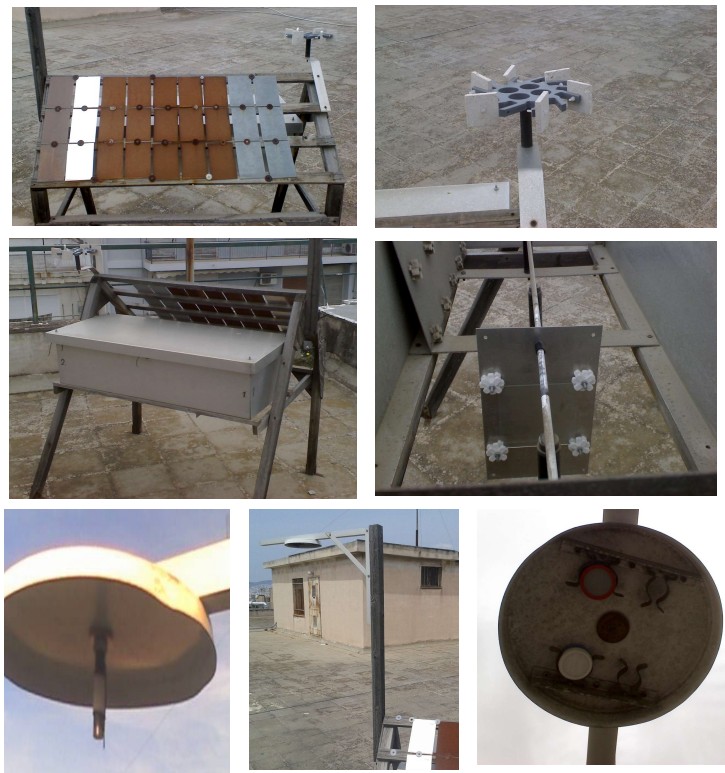

**Figure 1: The exposure site in the Athens centre (Greece). The top panel shows the carousel (on the right) and the main rack (on the left) with the material specimens, which was installed in Athens and consisted of an inclined plane and an aluminium box with open bottom (middle panel). The middle panel shows aluminium box (on the left) and the glass specimens in the aluminium box (on the right). The bottom panel shows the diffusive passive samplers for the surface air-pollutants measurements and the passive particle collector under the rain shield.**



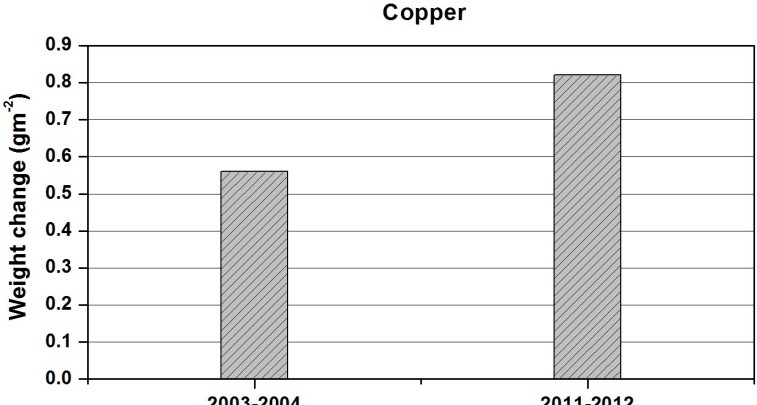

**Figure 2: Mean weight change of copper samples exposed during the periods 2003-2004 and 2011-2012.**



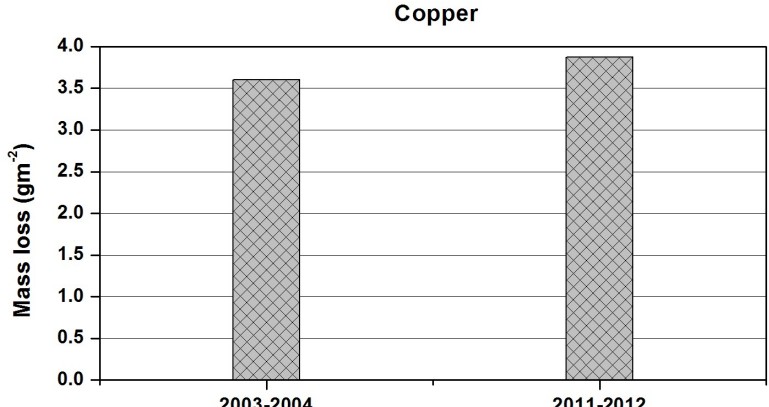

**Figure 3: Mean mass loss of copper samples exposed during the periods 2003-2004 and 2011-2012.**





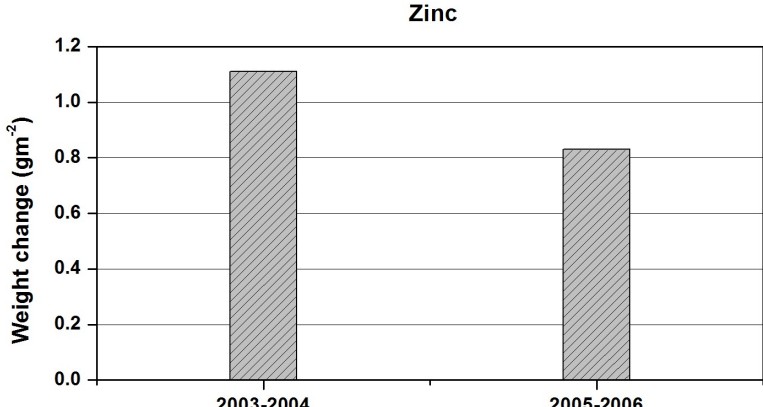

**Figure 4: Mean weight change of zinc samples exposed during the periods 2003-2004 and 2005-2006.**





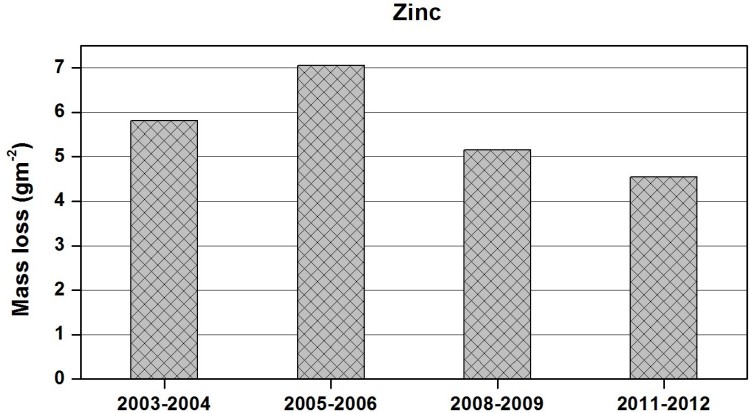

**Figure 5: Mean mass loss of zinc samples exposed during the periods 2003-2004, 2005-2006, 2008-2009 and 2011-2012.**





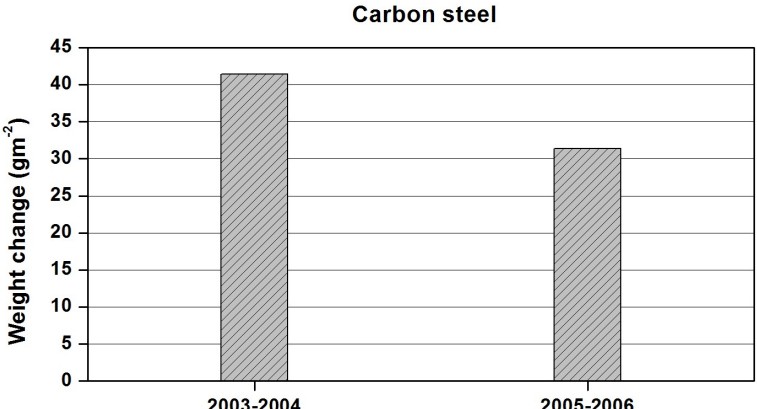

**Figure 6: Mean weight change of carbon steel samples exposed during the periods 2003-2004 and 2005-2006.**





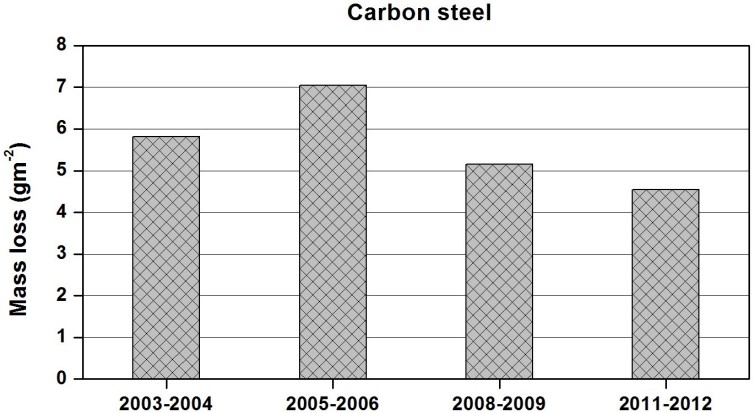

**Figure 7: Mean mass loss of carbon steel samples exposed during the periods 2003-2004, 2005-2006, 2008-2009 and 2011-2012.**



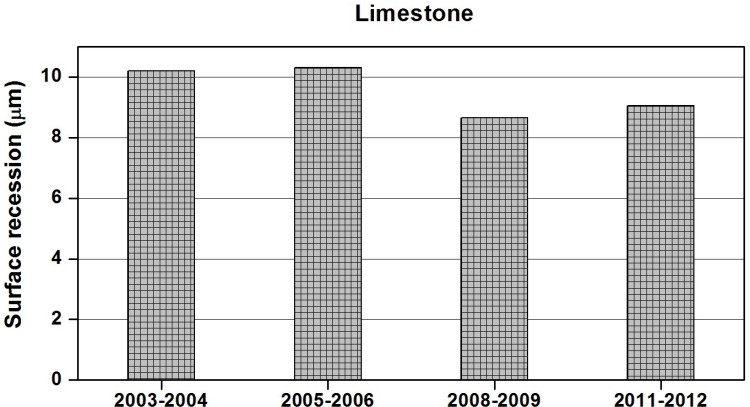

**Figure 8: Surface recession of limestone exposed in unsheltered positions for the periods 2003-2004, 2005-2006, 2008-2009 and 2011-2012.**





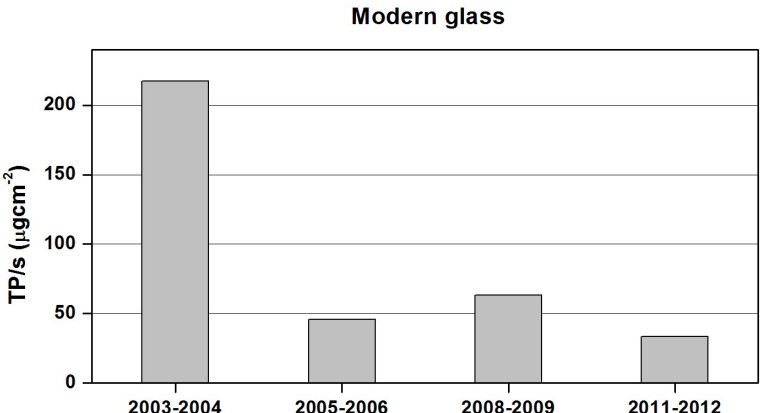

Figure 9: TP/S (µg cm$^{-2}$) for modern glass exposed for the periods 2003-2004, 2005-2006, 2008-2009 and 2011-2012.





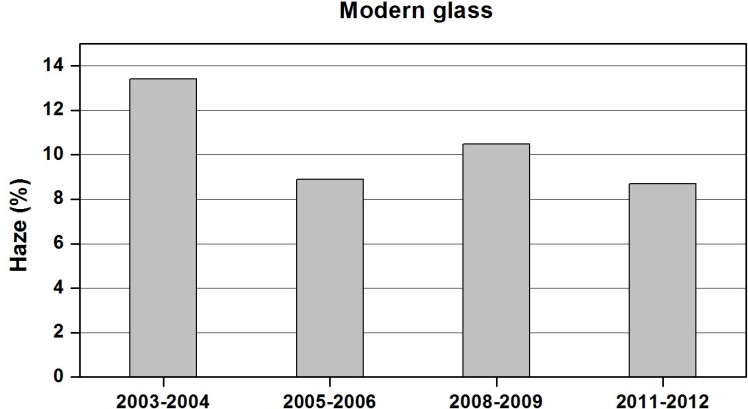

Figure 10: Haze (%) for modern glass exposed for the periods 2003-2004, 2005-2006, 2008-2009 and 2011-2012.





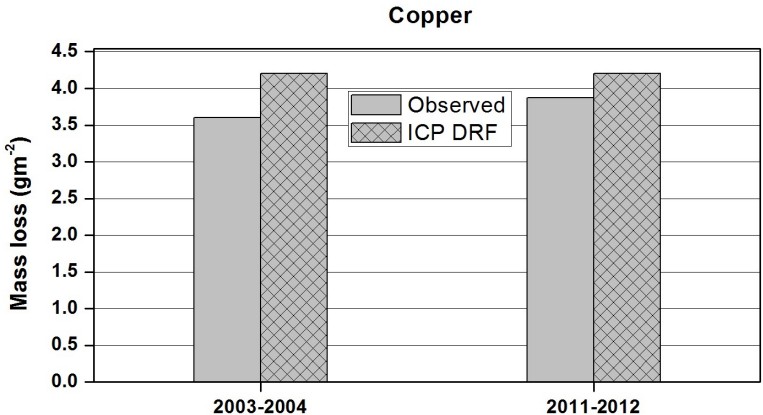

**Figure 11: Experimental obtained mass loss values at Athens, Greece for the case of copper along with the predicted ones by DRFs.**



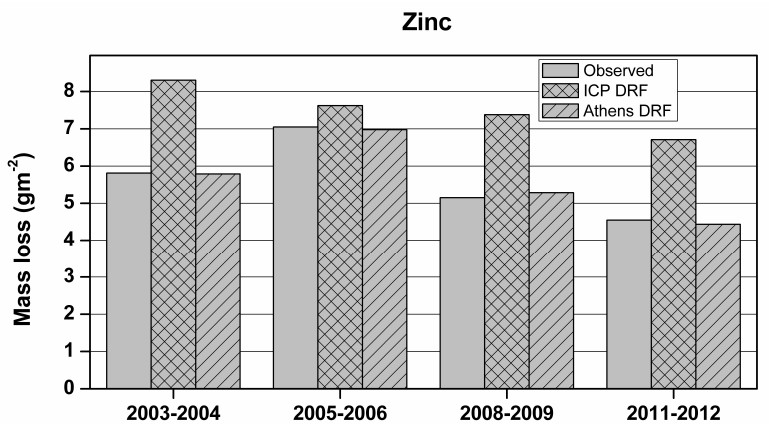

**Figure 12: Experimental obtained mass loss values at Athens, Greece for the case of zinc along with the predicted ones by DRFs.**



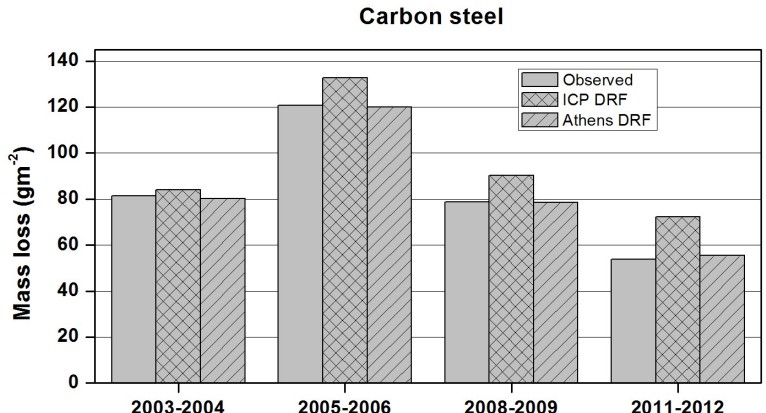

**Figure 13: Experimental obtained mass loss values at Athens, Greece for the case of carbon steel along with the predicted ones by DRFs.**





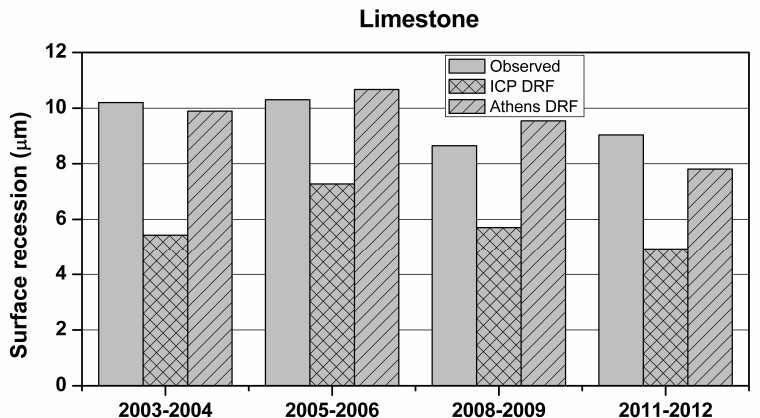

**Figure 14: Experimental obtained recession values at Athens, Greece for the case of limestone along with the predicted ones by DRFs.**





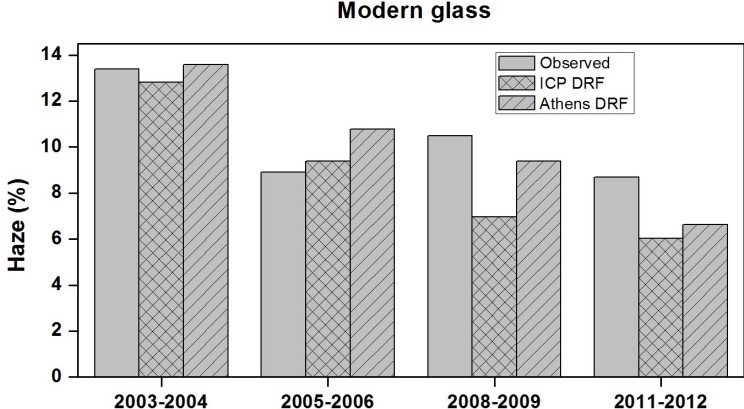

**Figure 15: Experimental obtained haze values at Athens, Greece for the case of modern glass along with the predicted**
5  **ones by DRFs.**



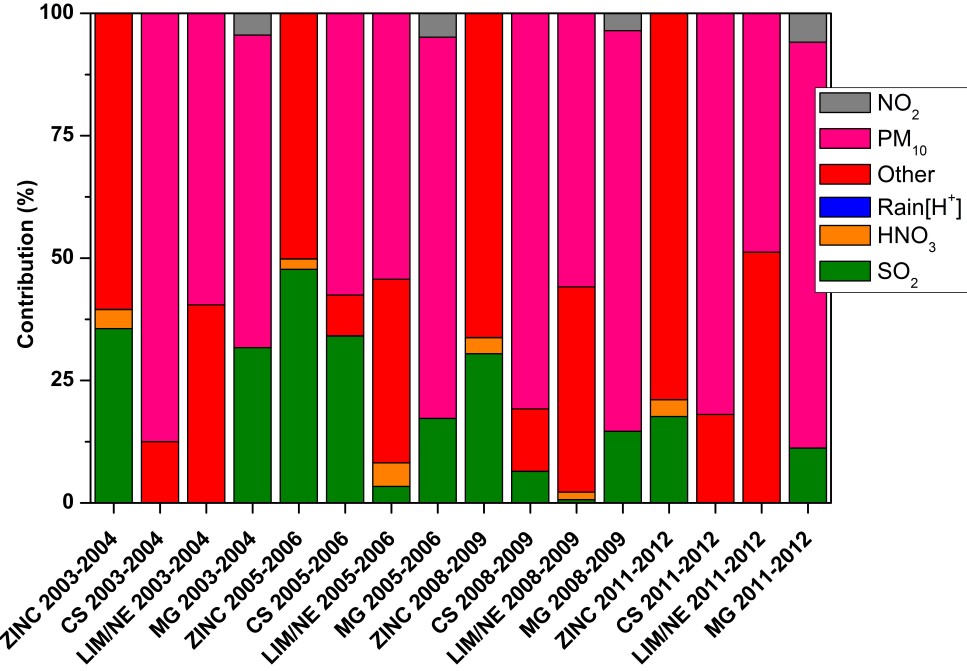

**Figure 16: The percentage contribution of each Athens DRF factor to the total corrosion/soiling of each material for all exposure periods. "CS" stands for Carbon Steel, "LIM/NE" stands for Limestone and "MG" stands for Modern Glass.**