# Peer review of "Impacts of air pollution and climate on materials in Athens, Greece"

_Atmospheric Chemistry and Physics, 2016_

## Referee Comment (RC1) · Anonymous Referee #1 · 27 Jun 2016

Review of the manuscript entitled "Impacts of air pollution and climate on materials in Athens, Greece" by J. Christodoulakis, C.G. Tzanis, C.A. Varotsos, M. Ferm, J. Tidblad

This paper presents and discusses corrosion/soiling experimental results of different materials (carbon and weathering steel, copper, zinc, limestone, modern glass) due to air pollution, together with climatic parameters, obtained during different one year exposure periods performed at Athens, Greece, since 2003. The authors also present/compare their results with corrosion/soiling estimations obtained using Dose Response Functions (DRFs for multi-pollutant situation) already presented in the literature and propose new DRFs targeted to Athens, Greece. The paper addresses relevant scientific questions within the scope of Atmospheric Chemistry and Physics journal. The overall presentation is also well structured and clear, and the conclusions are substantial. This manuscript is interesting because it presents new DRFs for different materials based on the new atmospheric multi-pollutant situation and climatic parameters at Athens, Greece. Therefore, I recommend publication of this paper after a few minor comments have been addressed. I would also like to notice that authors have taken into account all the comments made in my previous report.

Specific comments:

1. Page 1, line 13: Use capital for the initial letters of the words "dose response functions".

2. Page 1, line 14: "Dose" instead of "dose", "Response instead of "response".

3. Page 1, line 15: "Function" instead of "function".

4. Page 2, line 5: As before use capital for the initial letters of the words "dose response functions".

5. Page 2, line 26: Here it is referred that "sheltered samples" are exposed in the box under the rack while in the same page, line 29 is referred that only modern glass sample is exposed there. Please clarify.

6. Page 3, line 16: ". . . structural metals/alloys" instead of ". . . structural metals".

7. Page 3, line 20: ". . . structural metal/alloy" instead of ". . . structural metal".

8. I would suggest authors to unify figures where applicable, for example figs. 2 and 3, figs. 4 and 5, figs. 6 and 7, figs. 9 and 10.

9. Page 4, line 6: ". . . sensitive alloys" instead of "sensitive metals".

10. Page 5, line 7: Be consistent with the "Dose Response Function" term.

11. In each given equation, with a few exceptions (Eqs. 4, 6, 10), there is a constant factor, meaning that even in case all other factors were zero there will be corrosion on materials. Could you please give an explanation about this?

12. Page 6, lines 31-33: Give the meaning of these terms in the same way as for the

case of HNO3 in Page 7, line 1. Erase the terms "annual average".

13. Page 6, line 34: Erase the term "annual average".

14. Page 7, line 15: "DRF (Eq. 3) estimations" instead of "DRFs estimations".

15. Page 7, lines 18-23: Specify which equation (equation number) is considered for each material.

---

## Referee Comment (RC2) · Anonymous Referee #3 · 22 Aug 2016

The manuscript presents corrosion and soiling results of materials exposed under real environmental conditions as well as corrosion and soling estimations obtained using dose response functions. Among the materials studied are copper, carbon steel, weathering steel, zinc, modern glass and limestone. The experimental campaign covers the period from 2003 to 2012. An important contribution of this work is the development of new dose response functions for the particular case of Athens, Greece based on the current pollutant situation. Such kind of information is not available in the literature.

I believe that the paper is consistent with the fields of Atmospheric Chemistry and Physics journal. The paper is well structured and follows journal instructions. I recommend the publication of the paper after the proposed minor changes have been made.

[Figure]

Comments

1. The parameter "H" of the equations 6 and 10 are not defined. Add its definition in the given list.

2. In the title of Eq. 4 are given the chemical characteristics of the weathering steel. This information should be erased from this point and added in text where weathering steel is referred. Same info for the rest metal/alloys should be added.

3. Different figures concerning the same material, like for examples 2 and 3 but also others, could be presented as one figure defined (a) and (b).

4. In fig. 16, I would suggest the authors to change the order of the materials in axis x. It would be more useful for the reader the results of each material to be placed side by side in chronological order.

5. In the legends of figures 11-15 add the equations numbers of DRFs.

6. Page 22, caption: "by ICP DRF" instead of "by DRFs".

7. Page 25, caption: "surface recession" instead of "recession"

---

## Author Comment (AC1) · 8 Sep 2016

Dear Editor,

We thank both you and the referees for the fruitful reviews on our manuscript (acp-2016-196). Find please below a point-by-point Response to the referees' comments. As you will see, we have taken into account all comments, suggestions etc made by the referees and we intend to revise our manuscript, accordingly.

Thanking you once more

Yours sincerely

Prof. Costas Varotsos

[Figure]

Point-by-point Response to Referees' comments:

——Referee #1:

Referee comment

"This paper presents and discusses corrosion/soiling experimental results of different materials (carbon and weathering steel, copper, zinc, limestone, modern glass) due to air pollution, together with climatic parameters, obtained during different one year exposure periods performed at Athens, Greece, since 2003. The authors also present/compare their results with corrosion/soiling estimations obtained using Dose Response Functions (DRFs for multi-pollutant situation) already presented in the literature and propose new DRFs targeted to Athens, Greece. The paper addresses relevant scientific questions within the scope of Atmospheric Chemistry and Physics journal. The overall presentation is also well structured and clear, and the conclusions are substantial. This manuscript is interesting because it presents new DRFs for different materials based on the new atmospheric multi-pollutant situation and climatic parameters at Athens, Greece. Therefore, I recommend publication of this paper after a few minor comments have been addressed. I would also like to notice that authors have taken into account all the comments made in my previous report."

Response No comment

Referee comment "Page 1, line 13: Use capital for the initial letters of the words "dose Response functions"."

Response: We shall insert the proposed change in the revised manuscript.

Referee comment "Page 1, line 14: "Dose" instead of "dose", "Response instead of "Response"."

Response: We shall insert the proposed change in the revised manuscript.

Referee comment "Page 1, line 15: "Function" instead of "function"."

Response: We shall insert the proposed change in the revised manuscript.

Referee comment "Page 2, line 5: As before, use capital for the initial letters of the words "dose Response functions"."

Response: We shall insert the proposed change in the revised manuscript.

Referee comment "Page 2, line 26: Here it is referred that "sheltered samples" are exposed in the box under the rack while in same page, line 29 is referred that only modern glass sample is exposed there. Please clarify."

Response: In the revised manuscript we 'll change "sheltered samples" to "sheltered sample".

Referee comment "Page 3, line 16: "... structural metals/alloys" instead of "... structural metals"."

Response: We shall insert the proposed change in the revised manuscript.

Referee comment "Page 3, line 20: "... structural metal/alloy" instead of "... structural metal"."

Response: We shall insert the proposed change in the revised manuscript.

Referee comment "I would suggest authors to unify figures where applicable, for example figs. 2 and 3, figs. 4 and 5, figs. 6 and 7, figs. 9 and 10."

Response: In the revised manuscript we are going to unify the proposed figures according to Referee's instructions.

Referee comment "Page 4, line 6: "... sensitive alloys" instead of "sensitive metals"."
Response: We shall insert the proposed change in the revised manuscript.

Referee comment "Page 5, line 7: Be consistent with the "Dose Response Function" term."

Response: We shall insert the proposed change in the revised manuscript.

[Figure]

Referee comment "In each given equation, with a few exceptions (Eq. 4, 6, 10), there is a constant factor, meaning that even in case all other factors were 0 there will be corrosion on materials. Could you please give an explanation about this?"

Response: In the given equations the constants denote materials' corrosion due to other factors which are not included in the presented equations. Such two factors are, for example, sunlight and wind.

Referee comment "Page 6, lines 31-33: Give the meaning of these terms in the same way as for the case of HNO3 in Page 7, line 1. Erase the terms "annual average"."

Response: We shall insert the proposed change in the revised manuscript.

Referee comment "Page 6, line 34: Erase the term "annual average"."

Response: The term "annual average" will be deleted from the manuscript.

Referee comment "Page 7, line 15: "DRF (Eq. 3) estimations" instead of "DRFs estimations"."

Response: We shall insert the proposed change in the revised manuscript.

Referee comment Page 7, lines 18-23: Specify which equation (equation number) is considered for each material.

Response: We shall add equations numbers in the revised manuscript.

——————————————————————————————————————————

——Referee #3:

Referee comment

"The manuscript presents corrosion and soiling results of materials exposed under real environmental conditions as well as corrosion and soling estimations obtained using dose Response functions. Among the materials studied are copper, carbon steel, weathering steel, zinc, modern glass and limestone. The experimental campaign covers the period from 2003 to 2012. An important contribution of this work is the development of new dose Response functions for the particular case of Athens, Greece based on the current pollutant situation. Such kind of information is not available in the literature. I believe that the paper is consistent with the fields of Atmospheric Chemistry and Physics journal. The paper is well structured and follows journal instructions. I recommend the publication of the paper after the proposed minor changes have been made."

Response: No comment

Referee comment "The parameter "H" of the equations 6 and 10 are not defined. Add its definition in the given list."

Response: We shall add parameter "H" definition in the given list.

Referee comment "In the title of Eq. 4 are given the chemical characteristics of the weathering steel. This information should be erased from this point and added in text where weathering steel is referred. Same info for the rest metal/alloys should be added."

Response: Information about weathering steel will be erased from Eq. 4 and will be added to the appropriate position in text. Same information about the rest metal/alloys will be also added in the manuscript.

Referee comment "Different figures concerning the same material, like for examples 2 and 3 but also others, could be presented as one figure defined (a) and (b)."

Response: In the revised manuscript we are going to unify the proposed figures according to Referee's instructions.

Referee comment "In fig. 16, I would suggest the authors to change the order of the materials in axis x. It would be more useful for the reader the results of each material to be placed side by side in chronological order."

Response: We shall revise fig. 16 according to Referee's instructions.

Referee comment "In the legends of figures 11-15 add the equations numbers of DRFs."

Response: We shall revise fig. 11-15 legends according to Referee's instructions.

Referee comment "Page 22, caption: "by ICP DRF" instead of "by DRFs"."

Response: We shall insert the proposed change in the revised manuscript.

Referee comment Page 25, caption: "surface recession" instead of "recession"

Response: We shall insert the proposed change in the revised manuscript.
* * *

---

## Author Response (AR1)

**Authors' response to Referees' comments**

Dear Editor,

We thank both you and the referees for the fruitful reviews on our manuscript and their support to our findings (acp-2016-196).

As you will see, we have taken into account all comments, suggestions etc made by the referees and revised our manuscript, accordingly.

Please find below a point-by-point response to the referees' comments along with our actions. The marked-up version of our manuscript (using track changes) follows this letter.

Thanking you once more

Yours sincerely

Prof. Costas Varotsos

**Point-by-point response to Referees' comments:**
**Referee #1:**
**Referee comment**
"This paper presents and discusses corrosion/soiling experimental results of different materials (carbon and weathering steel, copper, zinc, limestone, modern glass) due to air pollution, together with climatic parameters, obtained during different one year exposure periods performed at Athens, Greece, since 2003. The authors also present/compare their results with corrosion/soiling estimations obtained using Dose Response Functions (DRFs for multi-pollutant situation) already presented in the literature and propose new DRFs targeted to Athens, Greece. The paper addresses relevant scientific questions within the scope of Atmospheric Chemistry and Physics journal. The overall presentation is also well structured and clear, and the conclusions are substantial. This manuscript is interesting because it presents new DRFs for different materials based on the new atmospheric multi-pollutant situation and climatic parameters at Athens, Greece. Therefore, I recommend publication of this paper after a few minor comments have been addressed. I would also like to notice that authors have taken into account all the comments made in my previous report."

**Response**
No comment

**Referee comment**
"Page 1, line 13: Use capital for the initial letters of the words "dose response functions"."

**Response**
We made the proposed change in the revised manuscript.

**Referee comment**
"Page 1, line 14: "Dose" instead of "dose", "Response instead of "response"."

**Response**
We made the proposed change in the revised manuscript.

**Referee comment**
"Page 1, line 15: "Function" instead of "function"."

**Response**
We made the proposed change in the revised manuscript.

**Referee comment**
"Page 2, line 5: As before, use capital for the initial letters of the words "dose response functions"."

**Response**
We made the proposed change in the revised manuscript.

**Referee comment**
"Page 2, line 26: Here it is referred that "sheltered samples" are exposed in the box under the rack while in same page, line 29 is referred that only modern glass sample is exposed there. Please clarify."

**Response**
In the revised manuscript we changed "sheltered samples" to "sheltered sample".

**Referee comment**
"Page 3, line 16: "… structural metals/alloys" instead of "… structural metals"."

**Response**
We made the proposed change in the revised manuscript.

**Referee comment**
"Page 3, line 20: "… structural metal/alloy" instead of "… structural metal"."

**Response**
We made the proposed change in the revised manuscript.

**Referee comment**

"I would suggest authors to unify figures where applicable, for example figs. 2 and 3, figs. 4 and 5, figs. 6 and 7, figs. 9 and 10."

**Response**

In the revised manuscript we unified the proposed figures according to Referee's instructions and we also changed figures numbering, in captions and in text, accordingly.

**Referee comment**

"Page 4, line 6: "… sensitive alloys" instead of "sensitive metals"."

**Response**

We made the proposed change in the revised manuscript.

**Referee comment**

"Page 5, line 7: Be consistent with the "Dose Response Function" term."

**Response**

We made the proposed change in the revised manuscript.

**Referee comment**

"In each given equation, with a few exceptions (Eq. 4, 6, 10), there is a constant factor, meaning that even in case all other factors were 0 there will be corrosion on materials. Could you please give an explanation about this?"

**Response**

In the given equations the constants denote materials' corrosion due to other factors which are not included in the presented equations. Such two factors are, for example, sunlight and wind.

**Referee comment**

"Page 6, lines 31-33: Give the meaning of these terms in the same way as for the case of $HNO_3$ in Page 7, line 1. Erase the terms "annual average"."

**Response**

We made the proposed change in the revised manuscript.

**Referee comment**

"Page 6, line 34: Erase the term "annual average"."

**Response**

The term "annual average" was deleted from the manuscript.

**Referee comment**
"Page 7, line 15: "DRF (Eq. 3) estimations" instead of "DRFs estimations"."

**Response**
We made the proposed change in the revised manuscript.

**Referee comment**
Page 7, lines 18-23: Specify which equation (equation number) is considered for each material.

**Response**
We added equations numbers in the revised manuscript.

**Referee #3:**

**Referee comment**
"The manuscript presents corrosion and soiling results of materials exposed under real environmental conditions as well as corrosion and soling estimations obtained using dose response functions. Among the materials studied are copper, carbon steel, weathering steel, zinc, modern glass and limestone. The experimental campaign covers the period from 2003 to 2012. An important contribution of this work is the development of new dose response functions for the particular case of Athens, Greece based on the current pollutant situation. Such kind of information is not available in the literature.
I believe that the paper is consistent with the fields of Atmospheric Chemistry and Physics journal. The paper is well structured and follows journal instructions. I recommend the publication of the paper after the proposed minor changes have been made."

**Response**
No comment

**Referee comment**
"The parameter "H" of the equations 6 and 10 are not defined. Add its definition in the given list."

**Response**
We added parameter "H" definition in the given list.

**Referee comment**
"In the title of Eq. 4 are given the chemical characteristics of the weathering steel. This information should be erased from this point and added in text where weathering steel is referred. Same info for the rest metal/alloys should be added."

**Response**

Information about weathering steel was erased from Eq. 4 and was added in page 2, line 35. Same information about the rest metal/alloys were also added in page 2, lines 34-37.

**Referee comment**

"Different figures concerning the same material, like for examples 2 and 3 but also others, could be presented as one figure defined (a) and (b)."

**Response**

In the revised manuscript we unified the proposed figures according to Referee's instructions. We also changed figures numbering, in captions and in text, accordingly.

**Referee comment**

"In fig. 16, I would suggest the authors to change the order of the materials in axis x. It would be more useful for the reader the results of each material to be placed side by side in chronological order."

**Response**

We revised fig. 16 according to Referee's instructions.

**Referee comment**

"In the legends of figures 11-15 add the equations numbers of DRFs."

**Response**

We revised fig. 11-15 legends, adding equations numbers, according to Referee's instructions.

**Referee comment**

"Page 22, caption: "by ICP DRF" instead of "by DRFs"."

**Response**

We made the proposed change in the revised manuscript.

**Referee comment**

Page 25, caption: "surface recession" instead of "recession"

**Response**

We made the proposed change in the revised manuscript.

**Authors' changes in manuscript**

We added a few lines about the interaction of air pollutants with aerosols and made small editing corrections for reader's convenience.

[revised manuscript text omitted]